# Prediction of Intraparenchymal Hemorrhage Progression and Neurologic Outcome in Traumatic Brain Injury Patients Using Radiomics Score and Clinical Parameters

**DOI:** 10.3390/diagnostics12071677

**Published:** 2022-07-10

**Authors:** Yun-Ju Shih, Yan-Lin Liu, Jeon-Hor Chen, Chung-Han Ho, Cheng-Chun Yang, Tai-Yuan Chen, Te-Chang Wu, Ching-Chung Ko, Jonathan T. Zhou, Yang Zhang, Min-Ying Su

**Affiliations:** 1Department of Medical Imaging, Chi Mei Medical Center, Tainan 710, Taiwan; jam3jeffrey@yahoo.com.tw (Y.-J.S.); vince7978@hotmail.com (C.-C.Y.); taiyuanc@mail.cjcu.edu.tw (T.-Y.C.); porthoswu@yahoo.com.tw (T.-C.W.); crazyboy0729@gmail.com (C.-C.K.); 2Department of Radiological Sciences, University of California, Irvine, CA 92868, USA; yanlin0917@gmail.com (Y.-L.L.); jonathanzhou6@gmail.com (J.T.Z.); yangz17@uci.edu (Y.Z.); msu@uci.edu (M.-Y.S.); 3Department of Radiology, E-Da Hospital/I-Shou University, Kaohsiung 824, Taiwan; 4Department of Medical Research, Chi Mei Medical Center, Tainan 710, Taiwan; ho.c.hank@gmail.com; 5Department of Information Management, Southern Taiwan University of Science and Technology, Tainan 710, Taiwan; 6Graduate Institute of Medical Sciences, Chang Jung Christian University, Tainan 711, Taiwan; 7Department of Medical Sciences Industry, Chang Jung Christian University, Tainan 711, Taiwan; 8Department of Health and Nutrition, Chia Nan University of Pharmacy and Science, Tainan 717, Taiwan; 9Department of Radiation Oncology, Rutgers-Cancer Institute of New Jersey, Robert Wood Johnson Medical School, New Brunswick, NJ 08903, USA; 10Department of Medical Imaging and Radiological Sciences, Kaohsiung Medical University, Kaohsiung 807, Taiwan

**Keywords:** cerebral hemorrhage, tomography, X-ray computed, machine learning, contusions, brain injuries, traumatic, radiomics

## Abstract

(1) Background: Radiomics analysis of spontaneous intracerebral hemorrhages on computed tomography (CT) images has been proven effective in predicting hematoma expansion and poor neurologic outcome. In contrast, there is limited evidence on its predictive abilities for traumatic intraparenchymal hemorrhage (IPH). (2) Methods: A retrospective analysis of 107 traumatic IPH patients was conducted. Among them, 45 patients (42.1%) showed hemorrhagic progression of contusion (HPC) and 51 patients (47.7%) had poor neurological outcome. The IPH on the initial CT was manually segmented for radiomics analysis. After feature extraction, selection and repeatability evaluation, several machine learning algorithms were used to derive radiomics scores (R-scores) for the prediction of HPC and poor neurologic outcome. (3) Results: The AUCs for R-scores alone to predict HPC and poor neurologic outcome were 0.76 and 0.81, respectively. Clinical parameters were used to build comparison models. For HPC prediction, variables including age, multiple IPH, subdural hemorrhage, Injury Severity Score (ISS), international normalized ratio (INR) and IPH volume taken together yielded an AUC of 0.74, which was significantly (*p* = 0.022) increased to 0.83 after incorporation of the R-score in a combined model. For poor neurologic outcome prediction, clinical variables of age, Glasgow Coma Scale, ISS, INR and IPH volume showed high predictability with an AUC of 0.92, and further incorporation of the R-score did not improve the AUC. (4) Conclusion: The results suggest that radiomics analysis of IPH lesions on initial CT images has the potential to predict HPC and poor neurologic outcome in traumatic IPH patients. The clinical and R-score combined model further improves the performance of HPC prediction.

## 1. Introduction

Traumatic brain injury (TBI), most commonly caused by unintentional falls and motor vehicle crashes, is a serious condition that results in neuropsychiatric impairment, disability and death [1]. The annual incidence of TBI is estimated at up to 939 per 100,000 people worldwide [2]. At the moment of blunt head injury, primary brain injury occurs when the brain impacts on the surrounding confines of the skull bone and dura, which further ruptures the neurons and glial cells [3]. The high kinetic energy also fractures the microvessels in the brain and causes extravasation, leading to the high-attenuating contusion hemorrhage observed on non-contrast computed tomography (CT) [4]. Patients with contusion hemorrhages, also called traumatic intraparenchymal hemorrhages (IPH), typically receive non-surgical management initially if there is no significant mass effect. However, hematomas could expand during hospitalization following a secondary brain injury process [5,6]. As well as dysfunctional hemostasis, one proposed mechanism is the activation of specificity protein 1, nuclear factor-ĸB and the upregulation of sulfonylurea receptor 1 on endothelial cells that received a lower kinetic energy [3]. A further increase in blood–brain barrier permeability aggravates vasogenic edema and causes oncotic cell death and capillary fragmentation that leads to hematoma expansion observed on subsequent CT images.

Hemorrhagic progression of contusion (HPC) is associated with poor neurological outcome and in-hospital death [5,7,8,9,10]. Some (12–24%) of the patients showing HPC require surgery in addition to standard medical care [9,11,12]. Although the criteria for HPC differ among studies, reported occurrence ranges from 16–75% [5,13,14], which is not uncommon. Therefore, prediction of the risk factors for HPC is important, not only to provide prognostic information but also to identify patients who may benefit from intensive monitoring and timely surgery. Multiple research articles have investigated the risk factors for HPC and poor neurological outcome based on clinical conditions, laboratory data and radiological findings on the initial brain CT scans [5]. The most commonly reported predictive parameters for HPC were concurrent subdural hemorrhage (SDH) [7,11,12,13,15,16,17], concurrent subarachnoid hemorrhage (SAH) [12,15,16,17], a larger IPH volume [9,11,12,15] and the presence of coagulopathy [10,14,17,18]. Integration of the clinical and radiological information showed c-statistics from 0.72 to 0.77 for HPC prediction in previous studies [16,19], which still leaves room for improvement.

CT images not only provide radiological information for visual interpretation but also contain quantitative semantic and agnostic features that can be extracted with radiomics tools. Machine learning models generated from selected radiomics features have been shown to correlate well with clinically important diagnostic or prognostic outcomes [20]. Focusing on spontaneous intracerebral hemorrhages, extensive studies have been performed, which demonstrated the good predictive capability of models derived from radiomics features for hematoma expansion [21,22,23,24,25] and poor neurological outcome [24,26,27,28,29,30]. Furthermore, increased c-statistics were observed after the radiomics features were combined with clinical and radiological parameters constantly.

Despite the good prognostic value of radiomics in spontaneous IPH being reported, research evidence on traumatic IPH is scarce [31,32]. The purpose of this study was to demonstrate the feasibility of radiomics to predict HPC and poor neurologic outcome in traumatic IPH patients. Firstly, the radiomics models were built from the IPH lesions shown on the initial CT examinations for prediction, and the results were compared to those predicted using conventional clinical parameters and the lesion volume information. Secondly, the radiomics score (R-score) was added to the clinical parameters to build the combined model, and the performance was compared to assess the added value of the radiomics score in the outcome prediction.

## 2. Materials and Methods

### 2.1. Patient Identification, Baseline Parameters, and Outcomes’ Definitions

This retrospective cohort analysis was conducted by the identification of patients from the TBI database in Chi Mei Medical Center and its branch hospitals, which consisted of patients entering the emergency room (ER) due to head injuries from 2015 to 2017. The institutional review board approved this study and waived the requirement for informed consent. After reviewing the medical charts and images, there were initially 1110 adult patients who had CT-documented intracranial hemorrhages associated with blunt head trauma. Among them, 756 patients had extra-axial hemorrhages only and were excluded from the analysis. IPH was found on CT images of 354 patients. A follow-up CT was ordered either due to neurological deterioration or routinely on the following days, based on the decisions of neurosurgeons. Patients without a follow-up CT scan within 96 h from the initial CT scan were excluded. Furthermore, patients showing midline shifts more than 5 mm on initial CT, brain herniation syndromes, surgical intervention between the initial and follow-up CT scans or cerebral aneurysms were also excluded. Finally, a total of 107 patients who received non-operative management after initial CT scans were eligible for analysis. The patient selection flow diagram is shown in Figure 1.

Baseline patient characteristics include the following parameters: age, sex, cause of head injury (falling or motor vehicle collision), single or multiple IPH lesions, the total volume of IPH on initial CT, concurrent extra-axial hemorrhages (epidural, subdural, subarachnoid and intraventricular), laboratory data (platelet count, international normalized ratio (INR) and activated partial thromboplastin time ratio), systolic blood pressure at ER, GCS (Glasgow Coma Scale) at ER, ISS (Injury Severity Score), comorbid conditions (hypertension and diabetes mellitus) and antiplatelet medications. No patients were on anticoagulants. The distribution of baseline parameters is detailed in Table 1 and Table 2.

The first outcome was determined by the occurrence of HPC or not. HPC was determined based on the two CT examinations, defined by a more than 30% relative volume increase or a more than 10 mL absolute volume increase on the follow-up CT compared to the initial CT in a patient, according to the criteria used in prior studies [11,14,16]. The second outcome was determined based on the Glasgow Outcome Scale (GOS) at three-month intervals, and further dichotomized to either being poor (1 to 3) or good (4 and 5).

### 2.2. CT Protocols

Multi-detector CT scanners (SOMATOM Definition AS, SOMATOM Sensation 64, and SOMATOM Emotion 16, Siemens Healthineers) were used for image acquisition. The standard brain CT was acquired with tube voltage and tube current between 80–120 kVp and 250–300 mAs, respectively. Image coverage was from the occipital bone to the vertex. CT scans were performed with dimension size from 512 × 512 × 28 to 512 × 512 × 46. The slice thickness ranged from 3.6 mm to 5.0 mm and the in-plane resolution varied from 0.38 × 0.38 mm^2^ to 0.49 × 0.49 mm^2^.

### 2.3. Image Segmentation

Segmentation was performed manually by tracing the hyperdense IPH region of interest (ROI) on every axial slice of the initial and follow-up CT scan with ImageJ software (National Institutes of Health). Efforts were made to ensure that segmented ROI did not include any nearby hyperdense regions such as bone, dura or extra-axial hemorrhages. A board-certified neuroradiologist (Reader A, 6 years of experience) performed the segmentation and the results were verified by a senior board-certified neuroradiologist (Reader C, 21 years of experience). To evaluate the intra-reader and inter-reader agreements on segmentation results, ROI of 30 cases were delineated again by Reader A with 2 months apart, and by another neuroradiologist (Reader B, 4 years of experience) independently. The Dice coefficient was used for comparison of the segmentation results.

### 2.4. Radiomics Features’ Extraction and Selection

To minimize the variability of extracted features, the Hounsfield units (HU) for each set of CT images were rescaled to the range from −1024 HU to 3071 HU. The nearly raw raster data format was converted from CT images and processed by MATLAB 2020 a (The MathWorks). Feature extraction was performed using PyRadiomics. The classes of features were selected from the PyRadiomics library, including the first-order statistics, the shape-based parameters and the second-order texture features of Grey-Level Co-Occurrence Matrix, Grey-Level Run Length Matrix, Grey-Level Size Zone Matrix and Grey-Level Difference Matrix. Finally, a total of 107 radiomics features were extracted from each lesion ROI. To identify the uncorrelated features with maximum relevance, feature selection was performed using support vector machine (SVM) with the Gaussian kernel. The intraclass correlation coefficients (ICC) for the selected radiomics features were calculated.

### 2.5. Radiomics Score and Performance Evaluation

The selected radiomics features were used to build R-score models for prediction of HPC and poor GOS outcome, by using two methods: SVM with the Gaussian kernel and random subspace *k*-nearest neighbors (KNN) classifiers. The choices of the adopted models were made after testing various SVM, KNN, decision trees and discriminant algorithms. We conducted a 10-fold cross-validation process to prevent overfitting, whereby 90% of cases were randomly selected as the training set and the remaining 10% as the testing set. This procedure was repeated ten times to obtain the average results. The prediction thresholds for HPC and poor GOS were both set at R-score of ≥0.5. The receiver operating characteristic (ROC) curves and areas under the ROC curves (AUC) were used to evaluate the performance of the created radiomics models. 

### 2.6. Building of Combined Clinical–Radiomics Model

We performed multiple logistic regression analyses based on the R-score and the clinical parameters for the prediction of HPC and poor neurologic outcome. Baseline parameters that showed higher AUCs individually were selected as variables for clinical model establishment. We evaluated the predictive performances of clinical variables alone, in combination with initial total IPH volumes, and further in combination with R-score. Clinical parameters were modeled as categorical variables. Initial total IPH volume and R-score were modeled as continuous variables. Comparison of AUCs was evaluated with DeLong’s test and *p*-values < 0.05 were considered significant. The overall analysis flowchart from segmentation, preprocessing, feature selection, model building and evaluation of their performances is shown in Figure 2.

## 3. Results

### 3.1. Baseline Patient Characteristics

As listed in Table 1, there were 45 patients (42.1%) showing HPC in our studied population. Patients showing HPC had significantly larger initial IPH volumes and more of them were in the ISS ≥ 25 group. As shown in Table 2, there were 51 patients (47.7%) with unfavorable outcomes of GOS 1 to 3. Older age, a falling injury, larger IPH volume, combined SAH, combined IVH, thrombocytopenia, increased INR, lower GCS, higher ISS and history of hypertension were significantly associated with the consequences of poor neurological outcome.

### 3.2. Repeatability of Segmentations

The average Dice coefficient was 0.857 for repeated segmentation of Reader A, and 0.773 for independent segmentation between the two readers. The selected radiomics features, and the intra- and inter-reader ICCs, are shown in the Appendix A. When using the definition of ICC between 0.4 and 0.59 as fair, 0.60 to 0.74 as good and above 0.75 as excellent [33], all selected radiomics features had excellent intra-reader agreements. For inter-reader agreements, seven features were excellent, one was good and one was fair.

### 3.3. Performance of Radiomics Score with Case Examples

The average AUC obtained from 10-fold cross-validation for R-score alone to predict HPC was 0.7638, and the AUC for R-score alone to predict poor neurologic outcome was 0.8067. Details on accuracy, sensitivity and specificity are shown in Table 3. Two case examples of true positive and true negative R-score predictions for HPC are depicted in Figure 3. Another two case examples of false positive and false negative R-score predictions for HPC are depicted in Figure 4.

### 3.4. Combined Clinical–Radiomics Model for Prediction of Hemorrhagic Progression

Selected clinical variables including age, multiple IPH, concurrent SDH, ISS and INR for prediction of HPC showed an AUC of 0.7133. A clinical–volume model, created by adding initial total IPH volume as a variable, raised the AUC to 0.7412 non-significantly (*p* = 0.237). However, the further combination of the R-score with the clinical–volume model significantly increased the AUC to 0.8315 (*p* = 0.022). Figure 5 shows the ROC curves for the clinical–volume, R-score and combination models.

### 3.5. Prediction of Poor Neurologic Outcome

The clinical–volume model, built from selected clinical parameters of age, GCS, ISS, INR and initial total IPH volume for the prediction of poor neurologic outcome showed an AUC of 0.9247, which was higher than the R-score alone. The combination of the R-score with the clinical–volume model showed a non-significant change in the AUC at 0.9503 (*p* = 0.095). Figure 6 shows the ROC curves for the clinical–volume and R-score models.

## 4. Discussion

In this study, we set out to investigate the role of radiomics features in predicting hematoma progression and poor neurologic outcome in TBI patients. Our main finding is that the R-score, derived from selected radiomics features and modeled through SVM and KNN, can classify traumatic IPH patients in developing HPC or having poor neurological outcome with AUCs of 0.76 and 0.81, respectively. A further combination of the clinical and IPH volume parameters significantly increased the predictive performance of HPC with an AUC of 0.83. The results demonstrate the predictive capability of radiomics for hematoma progression and poor outcome in the TBI setting.

Radiomics features are quantitative data obtained from medical images, including first-order statistics and shaped-based and texture-related feature classes. The mathematically-defined radiomics features represent the image characteristics of the ROI, and many of them cannot be discerned by visual interpretation. It has been shown to provide a signature of the lesion, and further modeling for diagnostic, prognostic or predictive purposes could be performed with machine learning algorithms based on the selected features [20]. Another approach is deep learning-based radiomics, which does not rely on mathematically-predefined features but various neural networks to directly identify imaging information that are relevant to clinical problems. As compared with feature-based radiomics, deep learning-based radiomics may be advantageous in terms of repeatability and reproducibility. However, it requires a larger dataset, which is often unavailable on a clinical basis. The interpretability of the result is also problematic since the method is perceived as a “black box” [34]. Nevertheless, there are efforts being made to overcome the shortcomings mentioned above. To ensure robustness of the selected features, we performed intra- and inter-reader analysis and demonstrated acceptable repeatability. For the choices of model-building algorithms, we conducted multiple tests using different machine learning algorithms to identify the best performing ones by using methods reported before [29].

The initial IPH volume is the simplest objective parameter that could be calculated from CT images. Other radiological features such as the black hole sign, swirl sign, heterogeneous density, blend sign, hypodensities, irregular shape and island sign are all known to be positively associated with hematoma expansion and poor neurological outcome [35]; however, the evaluations of these shape and density features are subjective. Therefore, the application of radiomics can extract more objective features that are not retrievable based on vision and manual measurements. It has consistently been shown to be effective for the prediction of hematoma expansion and poor neurologic outcome in spontaneous IPH patients [21,22,23,24,25,26,27,28,29,30]. As for traumatic IPH, we herein demonstrated its usefulness to further expand the limited evidence currently available [31,32].

Since the R-score only considers features of the IPH lesion on CT images alone, it is paramount to incorporate the results of other important clinical and radiological predictors. We selected pertinent variables generally recorded during clinical practice to avoid missing data and created a clinical–volume model for comparison. For HPC prediction, we found significant associations with higher ISS and larger volumes on univariate analyses. Multiple IPH, concurrent SDH, age and INR were also probable risk factors, although non-significant. A higher ISS indicates a more serious injury to a patient on its whole body, which could be indirectly linked to more blood loss with dysfunctional hemostasis and the activation of systemic proinflammatory biomarkers related to neuroinflammation [36], and thus is a reported risk factor for HPC [8,13]. A larger initial IPH volume, as well as multiple IPH and concurrent SDH, indicate greater severity of the initial head trauma, thus increasing the volume of susceptible brain tissue for the secondary injury process and leading to hemorrhagic progression [6]. Therefore, these factors are known to pose higher risks of HPC [5,7,9,11,12,13,15,16,17]. Aging increases vulnerable brain tissue due to weakness of the microvasculature and decreased cerebral blood flow [5,37], which further adds risk to HPC [13,15,16,17,19]. Elevation of INR suggests coagulopathy and is positively correlated with HPC [10,17,18]. Our clinical–volume prediction model yielded an AUC of 0.74, which is similar to the results from others (0.72–0.77) [16,19]. The combination of the R-score significantly increased the predictive model’s performance, demonstrating the favorable effect of adding radiomics features to predict HPC in traumatic IPH patients in a similar way for spontaneous IPH patients [21,23,24,25].

Although the radiomics score can predict poor neurologic outcome with a decent AUC of 0.81, we found it unhelpful to further augment the prediction of the clinical–volume model, which had already reached 0.92 in our studied population. Age, GCS and ISS were notable strong prognostic determinants. Increased odds for poor outcome after TBI in elderly patients were frequently reported [38,39]. Comorbid conditions, reduced physical reserves and medication usage were possible reasons for poor outcomes. A lower GCS, mainly reflecting severe brain injury, was also observed to be associated with poor functional recovery [38,39]. ISS was reported to be an independent predictor for poor outcomes, which has been linked to respiratory failure [40]. Although not selected as parameters in our predictive model, comorbid conditions of hypertension and diabetes mellitus were associated with higher percentages of poor neurologic outcome in our studied population. Hypertension causes defective cerebral autoregulation; therefore, the cerebral blood flow decreases (ischemia) or increases (hyperemia) abnormally even with small changes in arterial pressure and aggravates the vulnerability of brain tissue [41]. Diabetes mellitus contributes to hyperglycemia, which is a modifiable risk factor for poor neurologic outcome. The mechanism is not clearly understood but associated lactic acidosis, electrolyte disturbances and inflammation are possible causes [42]. An intensive glycemic control target shows a small but statistically significant reduction in the risk of poor neurological outcome [43]. In general, clinical factors remained more important in terms of outcome prediction in patients with TBI.

Some limitations are noted in this study. The retrospective analysis contains biases related to patient selection. The manual segmentation process, the consistency of which was verified among readers, is time-intensive and unrealistic to be incorporated into clinical workflows currently. We tried to apply an automatic segmentation tool to our dataset; however, the segmentation results for traumatic IPH were suboptimal and extensive adjustments were still required. In contrast to spontaneous IPH, the multiplicity and lower imaging contrast due to close locations to bone and extra-axial hemorrhages make traumatic IPH lesions more difficult to segment automatically. Inconsistent performances were reported by recent studies, showing a wide range of Dice coefficient results for automatic traumatic IPH segmentation [44,45,46]; however, development in this field is rapid. By using a finely-tuned automatic segmentation tool for traumatic IPH, larger numbers of images can be processed timely for radiomics analysis in the near future. Lastly, due to limited cases, we could not perform validation using an independent dataset; therefore, the generalizability of our results needs to be further investigated.

## 5. Conclusions

We demonstrate the feasibility of radiomics analysis of initial CT images for the prediction of HPC and poor GOS in traumatic IPH patients. The combination of the R-score with clinical and lesion volume parameters showed significantly better predictivity of HPC than clinical–volume information only. The results suggest that radiomics analysis of IPH lesions on initial CT images has the potential to predict the risk of progression and aid in clinical management for traumatic IPH patients. Nevertheless, well-designed prospective cohort studies or randomized controlled trials are still required to add evidence on the beneficial role of radiomics in the future.

## Figures and Tables

**Figure 1 diagnostics-12-01677-f001:**
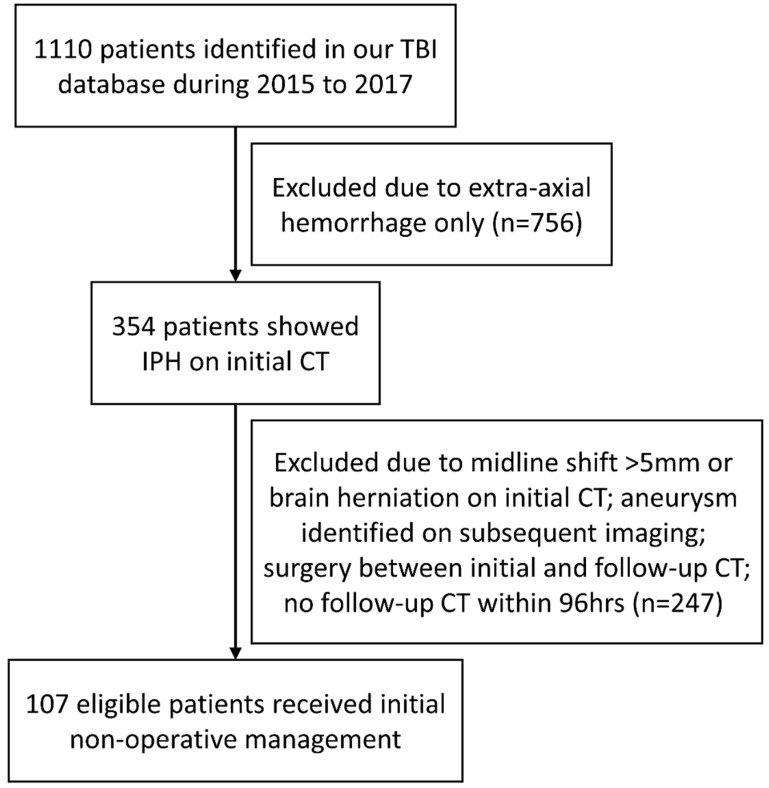
Flow diagram showing the patient identification processes. CT, computed tomography; IPH, intraparenchymal hemorrhage; TBI, traumatic brain injury.

**Figure 2 diagnostics-12-01677-f002:**
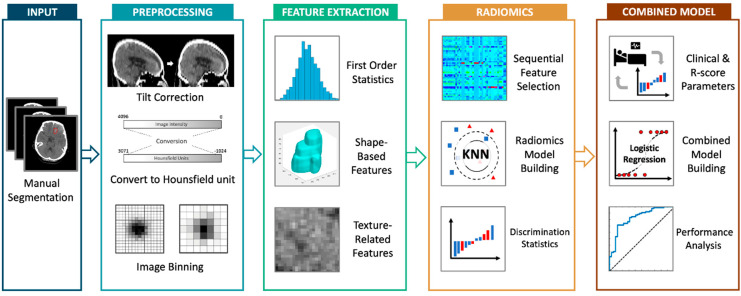
Overall steps for the establishment of progressive hematoma and poor neurological outcome prediction models. Following manual segmentation of intraparenchymal hemorrhage on initial CT images and preprocessing, radiomics features were extracted, selected and modeled through machine learning algorithms. The performance of radiomics scores and combined clinical–volume models were analyzed with receiver operating characteristic curves. R-score, radiomics score.

**Figure 3 diagnostics-12-01677-f003:**
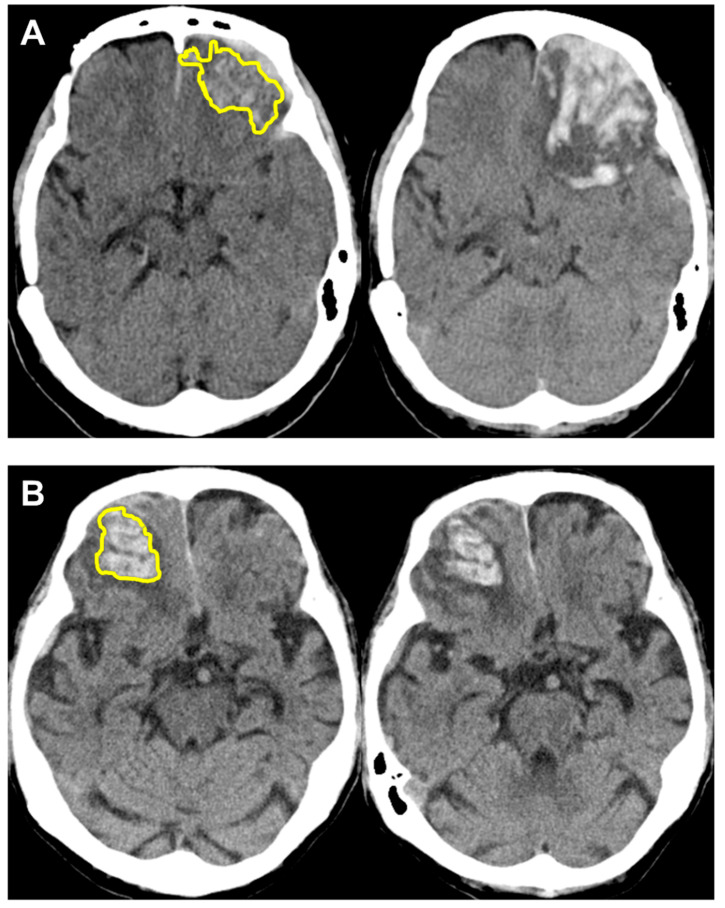
Case examples for successful R-score predictions of hemorrhagic progression, with initial images on the left and follow-up images on the right. R-scores were calculated based on selected radiomics features extracted from ROI segmentations shown in yellow. (**A**) True positive prediction (R-score 0.87) in a 51-year-old female with head injury due to scooter accident. A left frontal lobe contusion hemorrhage of 8.68 cc was noted initially. Deterioration of consciousness occurred 6 h later with GCS decreased from 15 to 9. Follow-up CT revealed progressive hematoma volume of 40.15 cc and she underwent craniectomy removal of the hematoma subsequently. (**B**) True negative prediction (R-score 0.25) in an 84-year-old female who sustained a falling injury to her head. The patient remained clinically stable and routine follow-up CT 64 h later revealed a stationary hematoma measuring 16.25 cc at the right frontal lobe. CT, computed tomography; GCS, Glasgow Coma Scale; R-score, radiomics score; ROI, region of interest.

**Figure 4 diagnostics-12-01677-f004:**
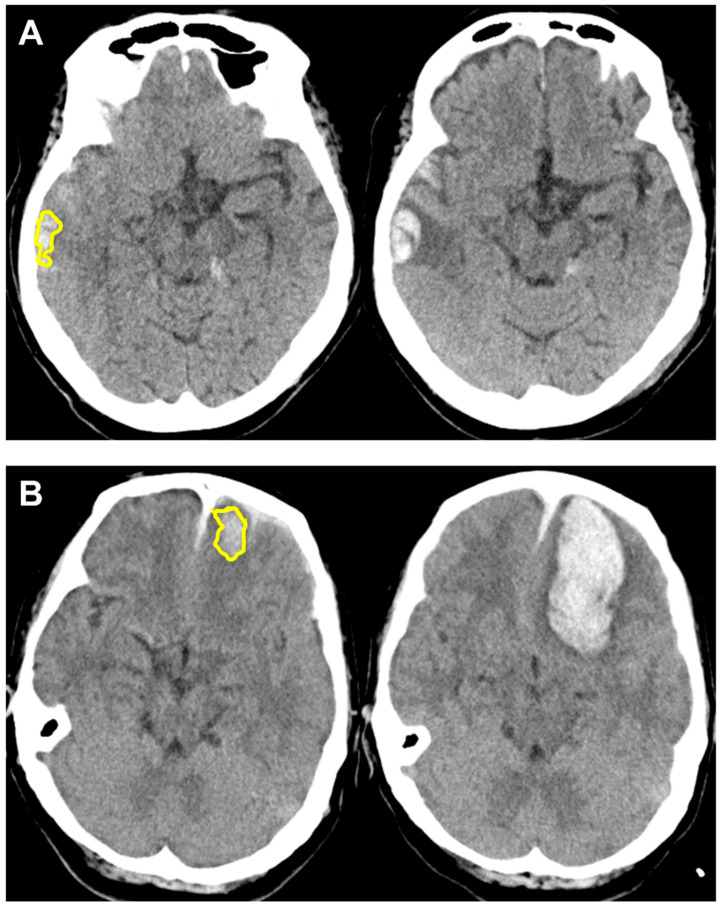
Case examples for unsuccessful R-score predictions of hemorrhagic progression, with initial images on the left and follow-up images on the right. R-scores were calculated based on selected radiomics features extracted from ROI segmentations shown in yellow. (**A**) False positive prediction (R-score 0.66) in an 84-year-old woman who had a head injury due to a scooter accident. The right temporal lobe hematoma measured 1.93 cc initially. After 48 h, the hematoma size was not significantly increased with a volume of 2.98 cc. (**B**) False negative prediction (R-score 0.39) in a 52-year-old woman who accidentally fell from height with an initial GCS of 14. The left frontal lobe hematoma expanded from 6.38 cc to 48.85 cc as her GCS decreased to 6 within 2 h. She underwent an emergent craniectomy removal of the hematoma. GCS, Glasgow Coma Scale; R-score, radiomics score; ROI, region of interest.

**Figure 5 diagnostics-12-01677-f005:**
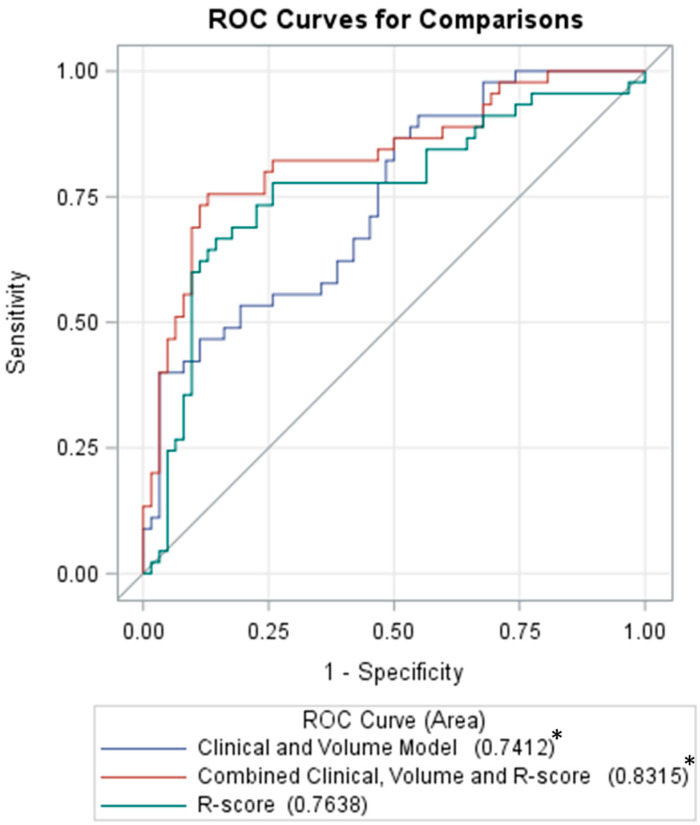
ROC curves of the clinical–volume, the R-score, and the combined models for the prediction of hemorrhagic progression. * Denotes significant difference between the two models with *p*-value < 0.05; R-score, radiomics score; ROC, receiver operating characteristic.

**Figure 6 diagnostics-12-01677-f006:**
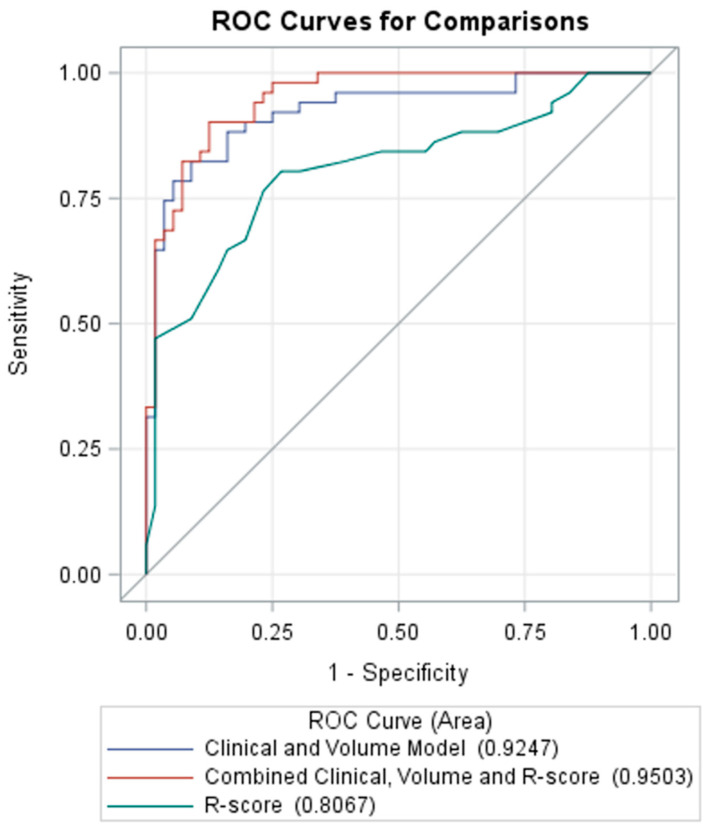
ROC curves of the clinical-volume, the R-score and the combined models for the prediction of poor neurologic outcome; R-score, radiomics score; ROC, receiver operating characteristic.

**Table 1 diagnostics-12-01677-t001:** Baseline characteristics of patients without or with HPC (*n* = 107).

	HPC		
	No (*n* = 62)	Yes (*n* = 45)	*p*-Value
Age (years), mean ± SD	52.82 ± 20.50	55.89 ± 18.31	0.426
<45	21 (33.87)	10 (22.22)	0.404
45–65	22 (35.48)	18 (40.00)	
≥65	19 (30.65)	17 (37.78)	
Sex, male (%)	42 (67.74)	32 (71.11)	0.833
Injury—Falling	25 (40.32)	18 (40.00)	1.000
Injury—Motor vehicle collision	37 (59.68)	27 (60.00)	
IPH volume (cc), mean ± SD	5.79 ± 8.19	11.32 ± 11.82	0.009 *
Multiple IPH, *n* (%)	37 (59.68)	34 (75.56)	0.100
EDH, *n* (%)	17 (27.42)	9 (20.00)	0.494
SDH, *n* (%)	41 (66.13)	37 (82.22)	0.080
SAH, *n* (%)	50 (80.65)	41 (91.11)	0.174
IVH, *n* (%)	8 (12.90)	5 (11.11)	1.000
PLT (count/μL), mean ± SD	239.52 ± 81.91	217.09 ± 56.28	0.096
<150 K	8 (12.90)	5 (11.11)	1.000
≥150 K	54 (87.10)	40 (88.89)	
Antiplatelet usage, *n* (%)	5 (8.06)	4 (8.89)	1.000
INR, mean ± SD	1.00 ± 0.13	1.02 ± 0.09	0.468
<1.05	53 (85.48)	31 (68.89)	0.056
≥1.05	9 (14.52)	14 (31.11)	
APTT ratio, mean ± SD	0.93 ± 0.11	0.93 ± 0.12	0.922
<1.05	54 (87.10)	39 (86.67)	1.000
≥1.05	8 (12.90)	6 (13.33)	
SBP at ER (mmHg), mean ± SD	149.65 ± 32.87	156.51 ± 34.25	0.297
<180	51 (82.26)	33 (73.33)	0.342
≥180	11 (17.74)	12 (26.67)	
GCS at ER, mean ± SD	12.23 ± 3.25	11.31 ± 3.65	0.175
<9	10 (16.13)	11 (24.44)	0.422
9–14	19 (30.65)	15 (33.33)	
≥14	33 (53.23)	19 (42.22)	
ISS score, mean ± SD	21.47 ± 10.45	25.24 ± 11.35	0.078
<16	9 (14.52)	4 (8.89)	0.009 *
16–25	33 (53.23)	13 (28.89)	
≥25	20 (32.26)	28 (62.22)	
HTN, *n* (%)	14 (22.58)	14 (31.11)	0.376
DM, *n* (%)	9 (14.52)	10 (22.22)	0.318

Categorical variables are analyzed by Fisher’s exact test and continuous variables are analyzed by independent samples *t*-test; * considered significant with *p*-value < 0.05; APTT, activated partial thromboplastin time; DM, diabetes mellitus; EDH, epidural hemorrhage; GCS, Glasgow Coma Scale; HPC, hemorrhagic progression of contusion; HTN, hypertension; IPH, intraparenchymal hemorrhage; ISS, Injury Severity Score; IVH, intraventricular hemorrhage; PLT, platelet; SAH, subarachnoid hemorrhage; SBP, systolic blood pressure; SD, standard deviation; SDH, subdural hemorrhage.

**Table 2 diagnostics-12-01677-t002:** Baseline characteristics of patients with good or poor neurologic outcome (*n* = 107).

	GOS		
	Good (4–5) (*n* = 56)	Poor (1–3) (*n* = 51)	*p*-Value
Age (years), mean ± SD	46.16 ± 17.90	62.84 ± 17.65	<0.001 *
<45	24 (42.86)	7 (13.73)	<0.001 *
45–65	25 (44.64)	15 (29.41)	
≥65	7 (12.50)	29 (56.86)	
Sex, male (%)	36 (64.29)	38 (74.51)	0.298
Injury—Falling	16 (28.57)	27 (52.94)	0.017 *
Injury—Motor vehicle collision	40 (71.43)	24 (47.06)	
IPH volume (cc), mean ± SD	6.15 ± 7.01	10.28 ± 12.56	0.042 *
Multiple IPH, *n* (%)	33 (58.93)	38 (74.51)	0.104
EDH, *n* (%)	16 (28.57)	10 (19.61)	0.368
SDH, *n* (%)	38 (67.86)	40 (78.43)	0.278
SAH, *n* (%)	42 (75.00)	49 (96.08)	0.002 *
IVH, *n* (%)	2 (3.57)	11 (21.57)	0.006 *
PLT (count/μL), mean ± SD	234.71 ± 65.50	225.00 ± 80.41	0.493
<150 K	3 (5.36)	10 (19.61)	0.036 *
≥150 K	53 (94.64)	41 (80.39)	
Antiplatelet usage, *n* (%)	2 (3.57)	7 (13.73)	0.083
INR, mean ± SD	0.98 ± 0.05	1.04 ± 0.15	0.006 *
<1.05	51 (91.07)	33 (64.71)	0.002 *
≥1.05	5 (8.93)	18 (35.29)	
APTT ratio, mean ± SD	0.90 ± 0.09	0.97 ± 0.13	0.004 *
<1.05	52 (92.86)	41 (80.39)	0.084
≥1.05	4 (7.14)	10 (19.61)	
SBP at ER (mmHg), mean ± SD	149.16 ± 26.70	156.24 ± 39.55	0.286
<180	47 (83.93)	37 (72.55)	0.166
≥180	9 (16.07)	14 (27.45)	
GCS at ER, mean ± SD	12.95 ± 2.56	10.63 ± 3.87	<0.001 *
<9	5 (8.93)	16 (31.37)	0.007 *
9–14	18 (32.14)	16 (31.37)	
≥14	33 (58.93)	19 (37.25)	
ISS score, mean ± SD	19.16 ± 7.31	27.33 ± 12.63	<0.001 *
<16	10 (17.86)	3 (5.88)	<0.001 *
16–25	32 (57.14)	14 (27.45)	
≥25	14 (25.00)	34 (66.67)	
HTN, *n* (%)	5 (8.93)	23 (45.10)	<0.001 *
DM, *n* (%)	7 (12.50)	12 (23.53)	0.205

Categorical variables are analyzed by Fisher’s exact test and continuous variables are analyzed by independent samples *t*-test; * considered significant with *p*-value < 0.05; APTT, activated partial thromboplastin time; DM, diabetes mellitus; EDH, epidural hemorrhage; GCS, Glasgow Coma Scale; HPC, hemorrhagic progression of contusion; HTN, hypertension; IPH, intraparenchymal hemorrhage; ISS, Injury Severity Score; IVH, intraventricular hemorrhage; PLT, platelet; SAH, subarachnoid hemorrhage; SBP, systolic blood pressure; SD, standard deviation; SDH, subdural hemorrhage.

**Table 3 diagnostics-12-01677-t003:** Performances of R-score on the predictions of hemorrhagic progression and poor neurologic outcome after 10-fold cross-validation.

	AUC	Accuracy	Sensitivity	Specificity
R-score for hemorrhagic progression	0.7638	0.7757	0.6222	0.8871
R-score for poor outcome (GOS 1–3)	0.8067	0.7664	0.7647	0.7679

AUC, area under the receiver operating characteristic; GOS, Glasgow Outcome Scale; R-score, radiomics score.

## Data Availability

The datasets generated or analyzed during the study are available from the corresponding author on reasonable request.

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
