# Peer review of "Prediction of Intraparenchymal Hemorrhage Progression and Neurologic Outcome in Traumatic Brain Injury Patients Using Radiomics Score and Clinical Parameters"

_diagnostics, 2022, doi:10.3390/diagnostics12071677_

Round 1

Reviewer 1 Report

Thanks interesting script and data. 

I think it is important to shed light on and discuss the strength and limitations of the various tests and technologies as a basis due to a general assessment.

However I think the conclusion should also state that prospective RCT studies are needed in future.

Reviewer 2 Report

The manuscript entitled “Prediction of Intraparenchymal Hemorrhage Progression and Neurologic Outcome in Traumatic Brain Injury Patients Using Radiomics Score and Clinical Parameters” is aimed to study if the combined model of radiomics score and clinical and brain volume data can better predict the incidence of Intraparenchymal Hemorrhage Progression and poor Neurologic Outcome in TBI Patients.

The authors have included the sufficient ROC curve and other data to support the conclusion. Two minor issues that can be addressed are

·        - Figure 6 is missing the ROC curve and value for the combined models.

    - Also, the authors can explain briefly if the high comorbidity % of patients especially that of diabetes incidence interfered with the observed results.
